# PHENAKI: VARIABLE LENGTH VIDEO GENERATION FROM OPEN DOMAIN TEXTUAL DESCRIPTIONS

**Ruben Villegas**[‡]
Google Brain
rubville@google.com

**Mohammad Babaeizadeh**[‡]
Google Brain
mbz@google.com

**Pieter-Jan Kindermans**[‡]
Google Brain
pikinder@google.com

**Hernan Moraldo**
Google Brain
hmoraldo@google.com

**Han Zhang**
Google Brain
zhanghan@google.com

**Mohammad Taghi Saffar**
Google Brain
msaffar@google.com

**Santiago Castro**[*]
University of Michigan
sacastro@umich.edu

**Julius Kunze**[*]
University College London
juliuskunze@gmail.com

**Dumitru Erhan**
Google Brain
dumitru@google.com

## ABSTRACT

We present Phenaki, a model capable of realistic video synthesis, given a sequence of textual prompts. Generating videos from text is particularly challenging due to the computational cost, limited quantities of high quality text-video data and variable length of videos. To address these issues, we introduce a new model for learning video representation which compresses the video to a small representation of discrete tokens. This tokenizer uses causal attention in time, which allows it to work with variable-length videos. To generate video tokens from text we are using a bidirectional masked transformer conditioned on pre-computed text tokens. The generated video tokens are subsequently de-tokenized to create the actual video. To address data issues, we demonstrate how joint training on a large corpus of image-text pairs as well as a smaller number of video-text examples can result in generalization beyond what is available in the video datasets. Compared to the previous video generation methods, Phenaki can generate arbitrary long videos conditioned on a sequence of prompts (i.e. time variable text or *a story*) in open domain. To the best of our knowledge, this is the first time a paper studies generating videos from open domain time variable prompts. In addition, compared to the per-frame baselines, the proposed video encoder-decoder computes fewer tokens per video but results in better spatio-temporal consistency.

## 1 INTRODUCTION

It is now possible to generate realistic high resolution images given a description [38, 39, 36, 42, 65], but generating high quality videos from text remains challenging. In essence, videos are just a sequence of images, but this does not mean that generating a long coherent video is easy. In practice, it is a significantly harder task because there is much less high quality data available and the computational requirements are much more severe [11]. For image generation, there are datasets with billions of image-text pairs (such as LAION-5B [45] and JFT4B [67]) while the text-video datasets are substantially smaller e.g. WebVid [4] with ∼10M videos, which is not enough given the higher complexity of open domain videos. As for computation, training current state-of-the-art image generation models is already pushing the state-of-the-art computational capabilities [65], leaving little to no room for generating videos, particularly videos of variable length.

To make the matters worse, one can argue that a single short text prompt is not sufficient to provide a complete description of a video (except for short clips), and instead, a generated video must be conditioned on a sequence of prompts, or *a story*, which narrates what happens over time. Ideally, a video generation model must be able to generate videos of arbitrary length, all the while having the capability of conditioning the generated frames at time $t$ on prompts at time $t$ that can vary over time. Such capability can clearly distinguish the *video* from a "moving image" and open up the way

---

[‡]Equal contribution. [*] Intern at Google Brain while working on this project.

**1st prompt:** "A photorealistic teddy bear is swimming in the ocean at San Francisco"

**2nd prompt:** "The teddy bear goes under water"

**3rd prompt:** "The teddy bear keeps swimming under the water with colorful fishes"

**4rd prompt:** "A **panda** bear is swimming under water"

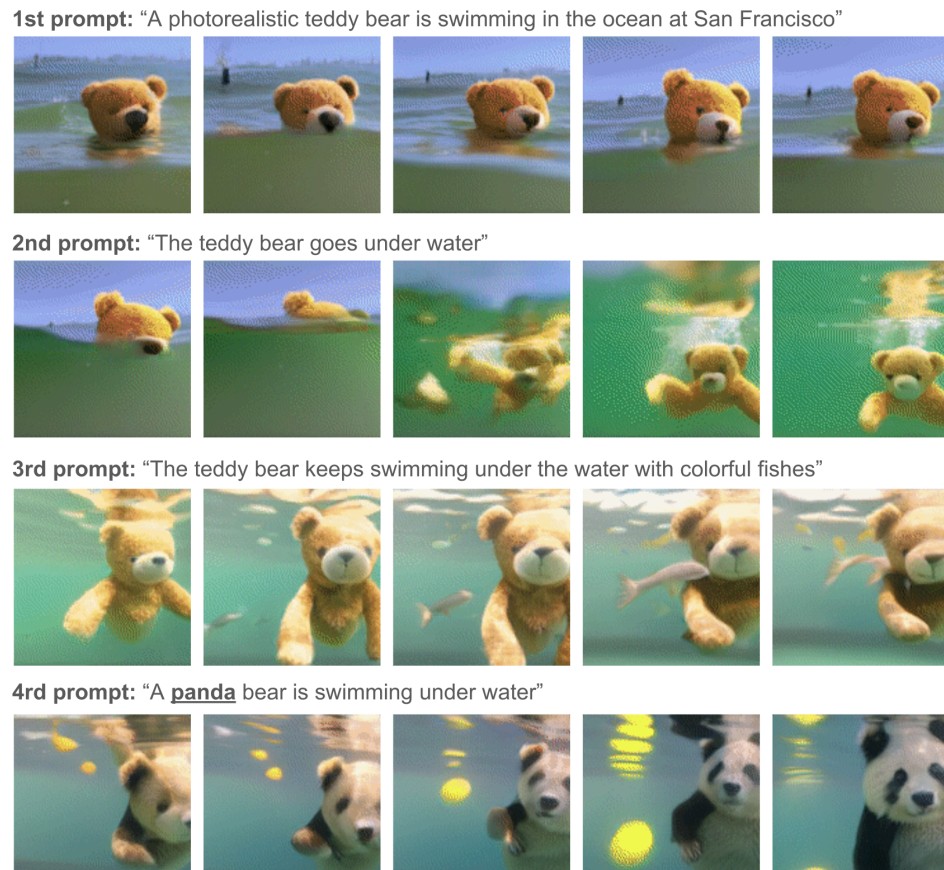

**Figure 1.** Time variable text (i.e. story) conditional video generation. The entire figure is **one continuous video** generated auto-regressively. We start by generating the video conditioned on the first prompt and then after a couple of frames we change the prompt to the next one. Each row contains a selected number of frames (from left to right in order) while the model was conditioned on that particular prompt. The model manages to preserve the temporal coherence of the video while adopting to the new prompt, usually taking the shortest path for the adaption (notice the *morphing* of the teddy bear to the panda). Note that the generated video has complex visual features such as reflections, occlusions, interactions and scene transitions. Full video is available at phenaki.github.io.

to real-world creative applications in art, design and content creation. To the best our knowledge, story based conditional video generation in open domain has never been explored before and this is the first paper to take early steps towards that goal. A traditional deep learning approach of simply learning this task from data is not possible, since there is no story-based dataset to learn from. Instead, to achieve this we rely on a model that is designed specifically with this capability in mind.

In this paper, we introduce Phenaki, a text to video model trained on both text to video and text to image data that can:

– Generate temporally coherent and diverse videos conditioned on open domain prompts even when the prompt is a new composition of concepts (Fig. 3). The videos can be long (minutes) even though the model is trained on 1.4 seconds videos (at 8 fps).

– Generate videos conditioned on a story (i.e. a sequence of prompts), e.g. Fig. 1 and Fig. 5.

To enable these capabilities, we could not rely on current video encoders, because they either can only decode fixed size videos or they encode frames independently. Hence, we introduce C-ViViT , a novel encoder-decoder architecture that:

– Exploits temporal redundancy in videos to improve reconstruction quality over a per frame model while compressing the number of video tokens by 40% or more.

– Allows encoding and decoding of variable length videos given its causal structure.

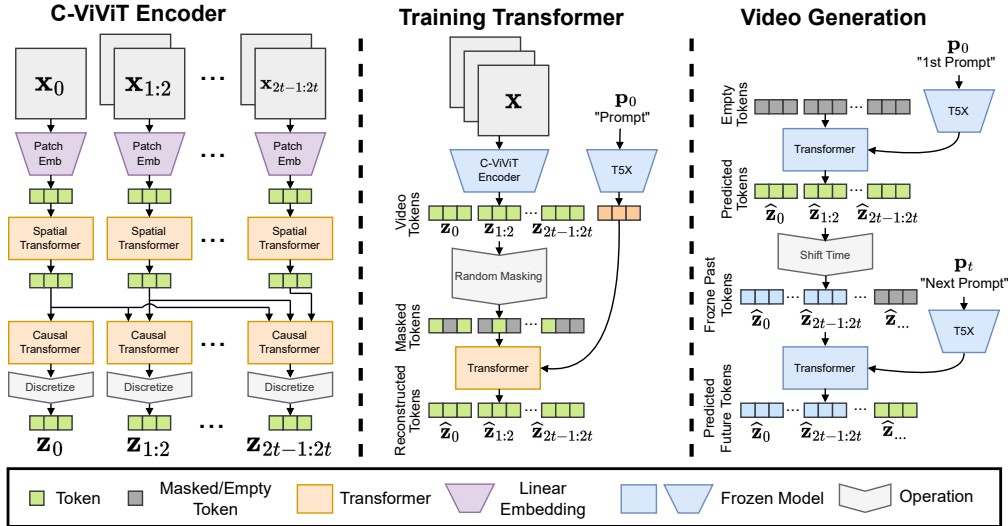

**Figure 2.** The architecture of Phenaki. **Left:** C-ViViT encoder architecture. The embeddings of images and video patches from raw frames **x** are processed by a spatial and then a causal transformer (auto-regressive in time) to generate video tokens **z**. **Center:** MaskGiT is trained to reconstruct masked tokens **z** predicted by a frozen C-ViViT encoder and conditioned on T5X tokens of a given prompt $\mathbf{p}_0$. **Right:** How Phenaki can generate arbitrary long videos by freezing the *past* token and generating the future tokens. The prompt can change over time to enable time-variable prompt (i.e. story) conditional generation. The subscripts represent time (i.e. frame number).

## 2 THE PHENAKI MODEL

Inspired by the previous work in auto-regressive text to image [38, 65, 42] and text to video [60, 59, 22], Phenaki is designed with two main components (see Figure 2): an encoder-decoder model which compresses videos to discrete embeddings (i.e. tokens) and a transformer model to *translate* text embeddings to video tokens. To get the text embeddings, Phenaki uses a pre-trained language model, T5-XXL [41]. We will discuss each one of these components in the following subsections.

### 2.1 ENCODER-DECODER VIDEO MODEL: C-ViViT

One of the primary challenges for generating video from text, is to get a compressed representation of videos. Previous work on text to video either use per-frame image encoders [22, 60, 63] such as VQ-GAN [14] or fixed length video encoders [58] such as VideoVQVAE [55]. The former allows for generating videos of arbitrary length, however in practice, the videos have to be short because the encoder does not compress the videos in time and the tokens are highly redundant in consecutive frames. The latter is more efficient in the number of tokens but it does not allow to generate variable length videos. In Phenaki, our goal is to generate videos of variable length while keeping the number of video tokens to a minimum so they can be modeled with a transformer within current computational limitations. To do so, we introduce C-ViViT, a causal variation of ViViT [1] with additional architectural changes for video generation, which can compress the videos in temporal and spatial dimensions, while staying auto-regressive in time. This capability allows for generating videos of arbitrary length auto-regressively.

**Encoder architecture:** As illustrated in Figure 2, we start with a video sequence of $t_x + 1$ frames with a resolution of $w_x \times h_x$ and $c_x$ channels: $\mathbf{x} \in \mathbb{R}^{(t_x+1) \times h_x \times w_x \times c_x}$. This sequence will be compressed into a token representation of size $(t_z + 1) \times w_z \times h_z$ where the first $w_z \times h_z$ tokens represent the first frame independently from the rest of the video, and the remaining tokens represent spatio-temporal video tokens that auto-regressively depend on previous frames. To do so, we extract non-overlapping image patches of size $w_p \times h_p \times c_p$ from the first frame and video patches of size $t_p \times w_p \times h_p \times c_p$ from the rest of the video. We typically use all channels at once such that the number of patches equals the number of video tokens $t_z = \frac{t_x}{t_p}$, $w_z = \frac{w_x}{w_p}$ and $h_z = \frac{h_x}{h_p}$. Each of these patches is flattened and linearly projected into a $d_z$ dimensional space. We combine the spatial

dimensions to have a tensor of shape $(t_z+1) \times w_z * h_z \times d_z$ where the spatial and temporal dimensions are separated. Then multiple transformer layers are applied along the spatial dimensions with all-to-all attention. This is followed by multiple transformer layers over the temporal dimension with causal attention such that each spatial token only observes spatial tokens from previous frames in an auto-regressive manner. The effect of this is that the first frame can be completely independently encoded. This opens up the possibility of text to image training to be embedded naturally into our video model. The second advantage is that we can condition the video generation process on a number of starting frames. The resulting patch embeddings $\mathbf{z}$ of shape $t_z \times w_z \times h_z \times d_z$ are then tokenized into learned codewords $c_z$ by vector quantization. The codebook learning will be discussed later together with the losses.

**Decoder architecture:** The C-ViViT decoder is simply an upside down version of the encoder. First tokens are transformed into embeddings. This is followed by the temporal transformer, then the spatial transformer. After the output of the spatial transformer, we apply a single linear projection without activation to map the tokens back to pixel space.

**Quantization and Losses:** To learn a discrete latent space, we quantize our encoder outputs into the entries of a learned codebook via the vector quantization (VQ) objective in VQVAEs [51],

$$L_{VQ} = \|\text{sg}(\mathbf{z}) - \mathbf{e}\|_2^2 + \beta\|\mathbf{z} - sg(\mathbf{e})\|_2^2, \tag{1}$$

where $\text{sg}(x) \equiv x$, and $\frac{\mathrm{d}}{\mathrm{d}x}\text{sg}(x) \equiv 0$ is the stop-gradient operator, $\beta$ is the commitment loss weight, and $\mathbf{e}$ is a codebook vector from codebook $\mathbf{E}$. The index to the codebook vector closest to $\mathbf{z}$ is found by $i = \text{argmin}_j\|\mathbf{z} - \mathbf{E}_j\|_2^2$. In addition to the VQ objective, we adopt the factorized and $\ell_2$-normalized codes from ViT-VQGAN [64] to improve codebook usage and reconstruction quality.

To train our model, we use a combination of $L_2$ loss, image perceptual loss $L_{IP}$ [24, 68], video perceptual loss $L_{VP}$ by using the I3D network [8] as feature extractor, and adversarial loss $L_{Adv}$ with StyleGAN architecture [25]. As training objective, we use the following

$$L = L_{VQ} + 0.1 \times L_{Adv} + 0.1 \times L_{IP} + 1.0 \times L_{VP} + 1.0 \times L_2. \tag{2}$$

**Novelty over the ViViT architecture:** While our proposed C-ViViT architecture is inspired by the factorized encoder in ViViT [1], we modify their architecture to enable self-supervised learning from unlabeled videos. We first remove the [CLS] tokens in the spatial and the temporal transformers. Next, we apply temporal transformer for all spatial tokens computed by the spatial encoder, in contrast to single run of the temporal transformer over the [CLS] tokens in ViViT. Most importantly, the ViViT encoder requires a fixed length video input due to the all-to-all attention in time. Therefore, we apply causal attention instead such that our C-ViViT encoder becomes auto-regressive and allows for a variable number of input frames which are necessary to learn from image datasets, and auto-regressively extrapolate video or single frames into the future.

## 2.2 TEXT-TO-VIDEO GENERATION WITH BIDIRECTIONAL TRANSFORMERS

In this stage, the text-to-video task can be formulated as a sequence-to-sequence problem to predict video tokens given the paired text embeddings. Most of recent methods [38, 65, 60, 22] adopt a transformer model for these sequence-to-sequence tasks. In their models, they use an auto-regressive transformer which predicts the image or video tokens sequentially given the encoded text features. As a result, the sampling time scales linearly with the sequence length, even when caching is used. This becomes impractical for long video sequence generation.

**Masked bidirectional transformer:** In this work, we aim to reduce the sampling time by having a small and fixed sampling step disregarding different video sequence lengths. Inspired by previous work for image generation [10], we use a bidirectional transformer since it can predict different video tokens simultaneously. For training step $i$, we first sample a mask ratio $\gamma_i$ from 0 to 1 and randomly replace $\lceil \gamma_i \cdot N \rceil$ tokens with the special token [MASK], where $N$ is the video sequence length. Then we learn the model parameters by minimizing the cross entropy loss on those masked tokens given the encoded text embeddings and unmasked video tokens. During inference, we first label all of the video tokens as the special token [MASK]. Then, at each inference step, we predict all the masked (unknown) video tokens in parallel conditioned on the text embeddings and unmasked

(predicted) video tokens. We keep a ratio $\beta_i$ of the predicted tokens at sampling step $i$ and the remaining tokens are re-masked and re-predicted in the next step.

As discussed in MaskGIT [10], the masking schedule $\gamma_i$ and sampling schedule $\beta_i$ have a significant effect on the samples quality therefore we follow the same strategies. Compared to an auto-regressive transformer, the number of sampling steps is an order-of-magnitude smaller (typically we use values in the range of 12 to 48). Generally speaking, more sampling steps improves the quality.

**Losses and training strategies:** Given a pre-trained C-ViViT , videos are encoded into codebook ids $\mathbf{a}$ of shape $(t_z + 1) \times w_z \times h_z$ which are flattened into a long vector using the raster ordering from [64]. We then model the text-conditional video token distribution using *Masked Visual Token Modeling* (MVTM) [10]:

$$L_{\text{mask}} = -\sum\nolimits_{\forall i \in [1,N], m_i = 1} \log p(a_i | \mathbf{a}_{\bar{M}}, \mathbf{p}), \tag{3}$$

where $\mathbf{a}_{\bar{M}}$ represents the masked version of $\mathbf{a}$, $m_i$ is a binary variable indicating whether $a_i$ is masked or not, $N$ is the number of video tokens, and $\mathbf{p}$ is the text condition embedding. In addition to the MVTM objective, we train using classifier-free guidance by dropping the text condition $10\%$ of the time during training [20, 65]. Finally, we dynamically adjust the MVTM objective during training to allow the use of image and video datasets as a single large dataset. We achieve this by only applying the masking ratio and objective on the first $w_z \times h_z$ tokens if only a single frame is given or over all video tokens if a full video is given. This mixed image and video dataset training strategy allows our models to learn concepts only present in image datasets, and transfer them to concepts present video datasets (e.g., the pencil drawing styled video of the panda in Figure 3).

**Inference and auto-regressive generation of long videos:** At inference time, we sample videos tokens by the same iterative process used in [10] with classifier-free guidance scale $\lambda$ to control alignment between the generation and the text condition. Once the first $t_x + 1$ frames are generated in latent space, we can extrapolate additional frames auto-regressively by re-encoding the last $K$ generated frames in the last generated video using C-ViViT , initializing MaskGIT with the tokens computed by our C-ViViT encoder, and proceed to generate the remaining video tokens conditioned on a text input. During video extrapolation, the text condition can be the same or a different one which enables our model to dynamically create visual transitions between the previous and current text condition visual content, effective generating a visual story an described by the input text. Phenaki does not perform generation purely in latent space due to the design choices we made to take advantage of text-image and text-video datasets during training. The first $w_z \times h_z$ tokens must be single image (space only) tokens and the next $t_z \times w_z \times h_z$ tokens must be video (space-time) tokens. Therefore, MaskGIT will always expect this input arrangement during generation.

## 3 EXPERIMENTS

To evaluate Phenaki, we test it on the following tasks: 1) text conditional video generation, 2) text-image conditional video generation, 3) open domain time variable text conditional video generation (i.e.) story mode, 4) video quantization and 5) image conditional video generation a.k.a. video prediction. To the best of our knowledge, 3) open domain time variable text conditional video generation has not been explored in prior work. Given the dynamic nature of videos, we highly encourage readers to visit phenaki.github.io to check the generated videos. The website also includes qualitative comparisons to a subset of the prompts from the CogVideo paper [22]. While the focus is on the text to video generation tasks, it is remarkable that Phenaki is still competitive on the more traditional video tasks despite not being developed explicitly for these tasks.

### 3.1 TEXT CONDITIONAL VIDEO GENERATION

Currently there is no established benchmark for evaluating text to video methods. This makes comparing Phenaki to recent methods such as NUWA [60], CogVideo [22], NUWA-Infinity [59] and video diffusion models [21] difficult.

Unless specified otherwise, we train a 1.8B parameter Phenaki model on a corpus of $\sim$15M text-video pairs at 8 FPS mixed with $\sim$50M text-images plus $\sim$400M pairs of LAION-400M [45] (more

**Table 1.** Text to video comparisons on Kinetics-400 [26].

| Method | FID Image ↓ | FID Video ↓ |
|---|---|---|
| T2V [29] | 82.13 | 14.65 |
| SC [5] | 33.51 | 7.34 |
| TFGAN [5] | 31.76 | 7.19 |
| NUWA | 28.46 | 7.05 |
| Phenaki [0-Shot] | 37.74 | 3.84 |

**Table 2.** Text to video and text to image results highlighting the importance of image datasets in video models. Text-to-image evaluation is done on ∼40K images of LAION-400M [45].

| Data Split | Text to Video | | | Text to Image | |
|---|---|---|---|---|---|
| Vid% / Img% | CLIP ↑ | FID ↓ | FVD ↓ | CLIP ↑ | FID ↓ |
| 100% / 0% | 0.298 | 19.2 | 168.9 | 0.240 | 53.9 |
| 80% / 20% | 0.303 | 21.4 | 198.4 | 0.289 | 29.4 |
| 50% / 50% | 0.302 | 21.4 | 239.7 | 0.287 | 30.5 |

details in Appendix B.3). The model used in the visualisations in this paper was trained for 1 million steps at a batch size of 512, which took less than 5 days. In this setup 80% of the training data came from the video dataset and each image dataset contributed 10%.

**Qualitative evaluation:** Samples from this model can be seen in Figure 3 and additional samples are provided at phenaki.github.io. We observe that there is a high degree of control over both the actors and the background dynamics in the videos. The appearance of the actors and the video style can be adjusted by the text prompt as well (e.g. a regular video, a cartoon or a pencil drawing).

On phenaki.github.io we provide examples from prompts that were provided in the CogVideo [22] demo. Since there are substantial differences between these methods it is hard to compare them on an equal footing. As an example, there are massive differences in scale: 9B parameters for CogVideo and 1.8B for our model. Additionally, the training data is different. Finally, we do not know how representative the prompts in the CogVideo demo are for the general performance of the CogVideo.

**Quantative comparison:** The NUWA [60] paper provided a qualitative evaluation on Kinetics-400. Since the NUWA model is only 0.9B parameters we also use a model of the same size. Our model was trained on 50% video and 50% image data in this experiment. The NUWA model finetuned on Kinetics but the Phenaki model is not: it is evaluated in a *zero shot setting*. The results in Table 1 show that Phenaki achieves comparable generation quality, in a zero-shot setting, compared to previous text to video methods that were actually trained or finetuned on this dataset.

**On the importance of joint text-to-image and text-to-video training** While there are some text-video datasets, text-image datasets dominate the internet in terms of quality and quantity [34]. Consequently, there is simply not enough video data available to cover all the concepts present in text-image datasets. For example using only our video data, concepts such as pencil drawings or different painting styles cannot be learned. To be able to learn a model that can combine video dynamics with these additional concepts we have to combine training on image and video data. In Table 2, we evaluate the performance of using different ratios of video and images. We start with data splits of only video, and vary the ratio of image and video datasets up to using 50% image and 50% video datasets. In our results, we find that there is a trade-off in performance between models trained with only video video (i.e., significantly better FVD), and models trained with more image data (i.e., better text-video and text-image alignment, and significantly better FID in image datasets). On phenaki.github.io we show samples from different models side by side where this trade-off between control over the content and the quality of the dynamics can be seen. We believe that the trade-off between concepts and dynamics will be improved as the quality and size of text-video datasets increases in the future.

## 3.2 TEXT-IMAGE CONDITIONAL VIDEO GENERATION

Given that Phenaki can be conditioned on both still images and text, an interesting setup is to *animate* existing images given a text prompt. For this experiment, we use the same model from Section 3.1 but conditioned on unseen pictures (captured with our phones from local subjects) and a related prompt. As it can be seen in Figure 4 the model can generate coherent videos starting from the given images, while following the given prompts.

## 3.3 VISUAL STORY TELLING BY DYNAMIC TEXT INPUTS

A notable and useful feature of Phenaki is that it is auto-regressive in time. This allows for generating long videos, while the prompt changes over time. Time variable prompts can be thought of as a *story*;

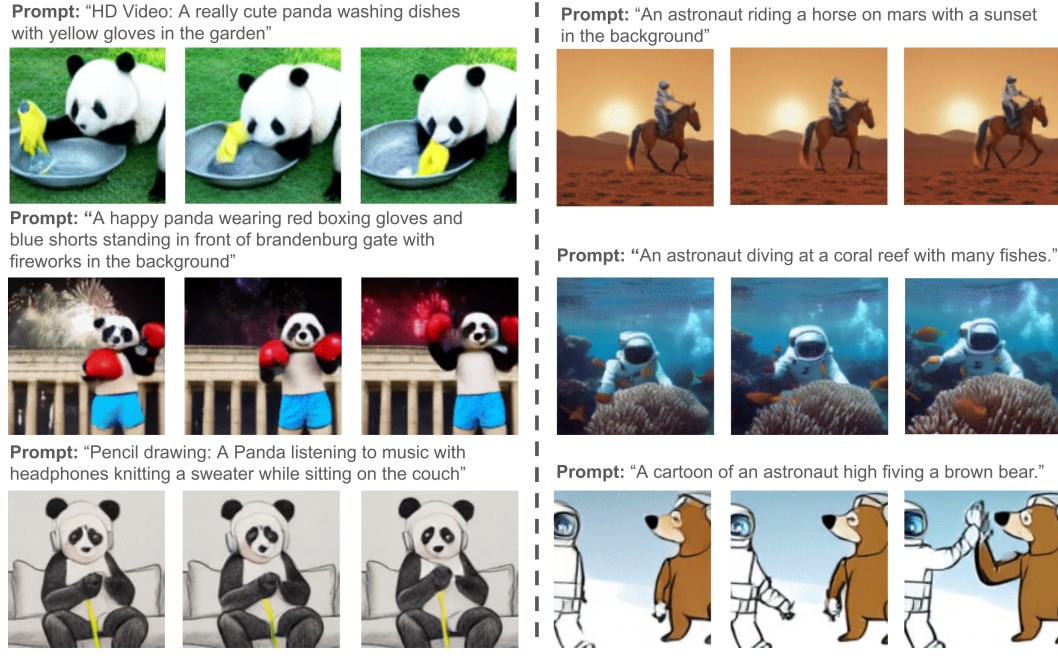

**Figure 3.** Text conditional video generation. Each row shows selected frames from a video generated given the prompt. The model is trained on a mix of images and videos. The video dataset does not include any *stylized* videos such as pencil drawings, however, the image dataset does. The model can generalize from still images to videos. This figure also demonstrate the capability of the model in generating new unseen compositions. Full videos are available at phenaki.github.io.

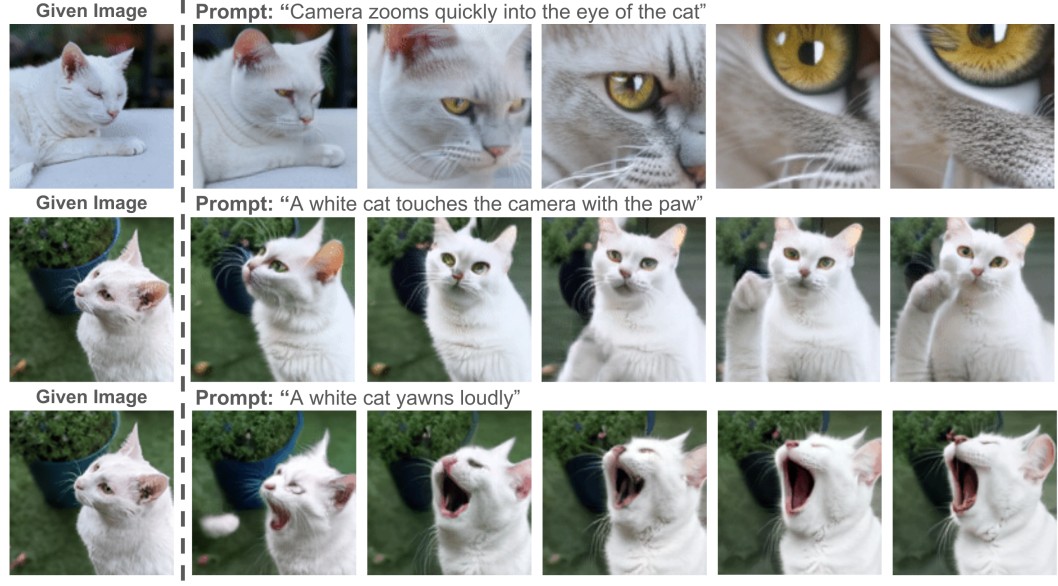

**Figure 4.** Animating images conditioned on a prompt. Each row demonstrates multiple frames of a generated video conditioned on a given first frame as well as a given text prompt. The first frames are new (captured by author's phone) and not observed during the training. The model *animates* the given image while following the prompt. Full videos are available at phenaki.github.io.

a narration of the entire video where each prompt corresponds to a scene from the video. This allows for creating dynamically changing scenes. To the best our knowledge, this paper is the first work to generate such videos. An example of this can be seen in Fig. 1 and on phenaki.github.io. The way it works is that we generate a video with the first prompt and then extend it in time by conditioning a possibly new prompt and on the last $N$, typically 5, previously generated frames.

**Table 3.** Video reconstruction results on Moments-in-Time. The number of tokens is computed for 10 frames with the exception of C-ViViT which is for 11, due to the isolated initial frame.

| Method | FID ↓ | FVD ↓ | Number of Tokens ↓ |
|---|---|---|---|
| Conv VQ-GAN [14] | 7.5 | 306.1 | 2560 |
| Conv VQ-GAN + Video loss | 13.7 | 346.5 | 2560 |
| ViT VQ-GAN [64] | 3.4 | 166.6 | 2560 |
| ViT VQ-GAN + Video loss | 3.8 | 173.1 | 2560 |
| C-ViViT VQ-GAN (Ours) | 4.5 | 65.78 | 1536 |

**Table 4.** Video prediction on Kinetics-600 [9]. While Phenaki is not designed for video prediction it achieves comparable results with SOTA video prediction models.

| Method | FVD ↓ |
|---|---|
| Video Transformer [57] | $170.0 \pm 5.00$ |
| CogVideo [22] | 109.2 |
| DVD-GAN-FP [11] | $69.1 \pm 0.78$ |
| Video VQ-VAE [55] | $64.3 \pm 2.04$ |
| CCVS [32] | $55.0 \pm 1.00$ |
| TrIVD-GAN-FP [31] | $25.7 \pm 0.66$ |
| Transframer [35] | 25.4 |
| RaMViD [23] | 16.5 |
| Video Diffusion [21] | $16.2 \pm 0.34$ |
| Phenaki (Ours) | $36.4 \pm 0.19$ |

**Table 5.** Video prediction on BAIR [13].

| Method | FVD ↓ |
|---|---|
| DVD-GAN [11] | 109.8 |
| VideoGPT [61] | 103.3 |
| TrIVD-GAN [31] | 103.3 |
| Transframer [35] | 100.0 |
| HARP [63] | 99.3 |
| CCVS [32] | 99.0 |
| Video Transformer [57] | 94.0 |
| FitVid [3] | 93.6 |
| MCVD [53] | 89.5 |
| NUWA [60] | 86.9 |
| RaMViD [23] | 84.2 |
| Phenaki (Ours) | 97.0 |

## 3.4 VIDEO ENCODING

To evaluate the video encoding and reconstruction performance of C-ViViT , we use the Moments-in-Time (MiT) [33] dataset. MiT contains ∼802K training, ∼33K validation and ∼67K test videos at 25 FPS. The MiT dataset, in contrast to other publicly available video datasets, is a high quality balanced dataset with high coverage and density of verbs depicting moments of a few seconds [33]. We compare C-ViViT against per-frame image based encoder-decoders that have been used as video quantizers for conditional video generation [63, 60, 22, 60, 22, 58]: a ViT [64] and a convolutional VQ-GAN[14]. The experimental details can be found in the Appendix B.1.

As demonstrated in Table 3, we evaluate the video reconstruction quality using FID [19] and FVD [50]. Both FID and FVD compare the distribution of generated videos (or images) to the ground truth distribution. The FID ignores temporal coherency, while the FVD measures how well the spatio-temporal dynamics of the videos are reconstructed. Results in Table 3 show that per-frame image based methods slightly outperform our video method (indicated by marginally higher FID of C-ViViT ), however, they do poorly at modeling the spatio-temporal dynamics in video (significantly lower FVD of C-ViViT ). This is expected as C-ViViT has spatio-temporal connections between patches in each frame, allowing space and time to be modeled together. In addition, C-ViViT compresses the video into fewer tokens per video compared to the image based baselines. This is crucial as the number of tokens drastically impacts the computational cost of the transformer in downstream tasks. Furthermore, C-ViViT tokens are auto-regressive in time which enables variable length videos to be modeled with the same encoder which is important for video extrapolation conditioned on previously generated frames.

## 3.5 IMAGE CONDITIONAL VIDEO GENERATION A.K.A VIDEO PREDICTION

To evaluate the learnt video representation of C-ViViT beyond reconstruction, we test it on the task of frame-conditioned video generation, also commonly known as video prediction [3]. In this experiment, we test Phenaki on BAIR Robot Pushing benchmark [13] where the task is to generate 15 frames conditioned on a given single frame. For open domain videos, we test Phenaki on Kinetics-600 [9] where the task is to predict 11 frames given 5 frames. More details about these experiments can be found in Appendix B.2. Tables 4 and 5 show the results of these experiments. Note that Phenaki is not specifically designed for video prediction, therefore, it lacks components such as skip connections in U-Nets which are known to improve the performance for video prediction methods [12, 52, 3]. Nevertheless, our method is competitive on these benchmarks with SOTA video prediction methods. Overall, these experiments show that Phenaki is strong at modeling dynamics of the videos which is required for generating coherent videos from text.

## 4 RELATED WORKS

This paper is closely related to auto-regressive methods for text conditioned image and video generation. DALL-E [38] *translates* text tokens to discrete image embeddings learnt using a VQVAE [51]. Parti [65] has a similar architecture but can generate higher quality images by predicting tokens from a ViT-VQGAN [64] using a 20B parameters transformer. Similar architectures have been used for generating videos as well. GODIVA [58] uses a transformer to map text tokens to video tokens from a image based VQVAE. Given the large number of tokens from multiple frames, GODIVA relied on a local-attention mechanism. Similarly, NUWA [60] and NUWA-Infinity [59] both employ auto-regressive architectures to generate videos and images from text. NUWA generates fixed size outputs, while NUWA-Infinity introduces a second layer of auto-regressive computation to support variable size videos. Likewise, CogVideo [22] argues the main reason behind low quality video generation is the scarcity of good text-video data and tried to leverage pre-trained text to images models to generate high quality video.

While Phenaki sticks to the same architecture principles, it has major differences with previous work. Most notably, NUWA, NUWA-Infinity and CogVideo treat videos as a sequence of independent images. This can lead to poor modeling of dynamics and generate motion artifacts. To combat this, NUWA-infinity used the previous frame during decoding to combat this. In Phenaki, we go further and treat videos as a temporal sequence of images which substantially decreases the number of video tokens given the redundancy in video generation, and results in a much lower training cost. The auto-regressive nature of the Phenaki also allows us to effectively condition on previous frames and generates longer videos as detailed in Section 2.

Diffusion models are another class of models which recently have been used for conditional and unconditional video generation, which we call VDM [21]. In VDM, authors proposed replacing the conventional U-Net architectures for 2D image modeling with a 3D space-time model to run the diffusion process directly on pixels. While this approach provides an effective formulation for modeling videos, it is limited to fixed size videos. To address this issue, VDM provides an auto-regressive extension, which allows the model to generate longer videos but it is typically impractical due to high sampling time of diffusion models.

Text conditional video generation is a relatively new field of research, nonetheless, image conditional video generation, commonly known as video prediction, and unconditional video generation have been studied more comprehensively. These papers include deterministic methods using a combination of recurrent and convolutional networks [40, 48, 15, 56], variational based stochastic methods [2, 12, 52, 3] and more recently by learning a discrete representation [55, 37, 35], auto-regressive models [57, 61, 32, 63], diffusion models [53, 18, 62, 23] flow based models [28], and finally adversarial based methods [54, 43, 49, 11, 44, 31, 66, 16, 46, 7, 17]. These works mostly consider limited domain (e.g. robotic videos, video games and small datasets) prediction/generation, or short fixed size clips. Section 3 and Section C provide comparison with some of these models.

## 5 CONCLUSION

We introduced Phenaki, a model which is capable of generating variable length videos conditioned on a sequence of open domain text prompts. Phenaki uses C-ViViT as video encoder. C-ViViT is a new model which provides temporal-spatial compression while being auto-regressive in time. The C-ViViT model is a crucial part of Phenaki that allows it to generate variable length videos. We demonstrate how joint training on images and videos can improve the generation quality, and diversity, given the existence of much larger image-text dataset with order of magnitude more samples. The Phenaki model achieves good performance on video prediction, it can be used as to generate long videos conditioned on a text prompt. Additionally it is able to condition on both text and a starting frame. Finally, Phenaki is not limited to generating a video depicting a single concept or caption. It is actually able to generate longer coherent video stories based on a sequence of text prompts. The more complex narratives it can visualize demonstrate how this can become a great creative tool for story telling.

## ETHICS STATEMENT

While we have not explored potential downstream applications of the generative models described in this work, we believe Phenaki can have a positive impact in a variety of creative settings. In general, many of the samples from the model will not perfectly correspond to the input caption or the user's intent; however, the end-user is likely to gain considerable time savings even if only one of the generated samples aligns with their intent. We thus foresee Phenaki being useful in eventually empowering users to accelerate their creativity, especially since the model can so quickly generate videos. Phenaki and similar models will be part of an ever-broad toolset for artists and non-artists alike, providing new and exciting ways to express creativity.

The flip-side of this acceleration and ease-of-use is the potential for harmful impact, as with many of the prior or concurrent work in generative modeling. An easy-to-use system like Phenaki can be repurposed for generating maliciously fake content and enable spreading of such content much easier. While the quality of the videos generated by Phenaki is not yet indistinguishable from real videos, getting to that bar for a specific set of samples is within the realm of possibility, even today. This can be particularly harmful if Phenaki is to be used to generate videos of someone without their consent and knowledge.

Like DALLE-2 [39], Imagen [42], Parti [65] and others, Phenaki is trained on a collection of datasets that is known to encode a number of undesirable biases. LAION-400M [45] specifically has a variety of issues regarding violence, pornography, gore [6]. While our primary image and video datasets have minimal traits like this, we did incorporate LAION-400M into our training and observed better results. In a currently training version of Phenaki, we use a set of datasets that minimizes such problems.

Another potential issue when training generative models of any sorts is that public datasets of imagery and videos may contain examples whose ownership status is unclear or impossible to verify. The Phenaki model is not an exception, as part of its training set uses the LAION-400M set of images. There is an ongoing public debate about how generative modeling research should proceed in the presence of potentially copyrighted works in the training set. Given this debate, we are taking a conservative stance by not releasing the trained checkpoints.

Taken together, these issues contribute to our decision not to release the underlying models, code, data or interactive demo at this time. Before we can do that, we want to focus our efforts on better understanding of data, prompt and output filtering. We would also like to more explicitly measure the biases encoded in the outputs of Phenaki, so that we can further mitigate them actively, either in the data, models or pre/post-processing steps.

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
