# OpenReview forum: "Phenaki: Variable Length Video Generation from Open Domain Textual Descriptions"
_ICLR.cc/2023/Conference — ICLR 2023 poster_

### Official Review · Reviewer_Qmqe · 2022-10-23

**Confidence:** 4
**Correctness:** 2
**Technical Novelty And Significance:** 3
**Empirical Novelty And Significance:** 3
**Recommendation:** 6

**Clarity, Quality, Novelty And Reproducibility:**

The paper is generally well written. The quality is affected by the missing baselines and necessary experiments. The architecture and video results are novel.

**Strength And Weaknesses:**

[Strengths]

1. The idea to extend ViViT to C-ViViT for encoding videos and images into discrete tokens is reasonable and novel.
2. Adapting the MaskGiT-like framework to video VQGAN models is novel.
3. The ablation study of the image-video joint training is valuable to the community.
4. The quality of the generated short and long videos based on texts is much better than the previous arts.

[Weaknesses]

The paper misses an important baseline [1] which also shows results in generating videos based on variable prompts. [1] shares a similar goal of generating long videos with a similar video VQGAN architecture. Therefore, I think it is important to discuss and compare this method, especially on the following points:

1. [1] also leverages a VQGAN framework for video generation by extending the auto-encoder part with 3D convolutions. It would be necessary to compare with it in Table 3 on video reconstruction. Currently, only 2D convolution and per-frame ViT models are compared. It is not clear that C-ViViT works better than 3D convolution.
2. [1] also show video results conditioned on variable length of text prompts, which means the argument that this paper is the first to study this problem is not true. It is more reasonable to say this is the first to study generating videos from variable lengths of **open-domain** text prompts, since I believe [1] only shows results on a limited domain.

Including the above discussion should make the paper more complete and a stronger submission. In addition to the concern above, some of the experimental results in the paper are not convincing:

1. The authors claim that the video dataset does not contain any stylized videos like pencil drawings which makes such videos impossible to be learned solely from the video dataset. However, in the shown qualitative results, some difficult prompts like "astronaut riding a horse" can still be generated on Mars. Can the authors show examples of the "pencil drawing styles" with a model trained on video datasets only?

2. One advantage of using MaskGIT claimed in the paper is the speed of sampling longer sequences. However, there are no experiments showing that, and no ablation on the hypermeters $\gamma$, $\beta$, and sampling steps which are claimed to be important in the paper.

[1] Long video generation with time-agnostic vqgan and time-sensitive transformer. ECCV 2022.

**Summary Of The Paper:**

This paper studies text-to-video generation. It proposes an architecture based on the VQGAN framework. For the encoder-decoder part, a novel ViT-like model is introduced to encode both videos and images into discrete tokens, which allows joint training on the text-image dataset. For the sequence modeling part, a MaskGiT-like model is trained to predict video tokens. The authors show the effectiveness of the proposed framework as well as joint training on image / text conditional generation with qualitative and quantitative evaluation.

**Summary Of The Review:**

This paper shows improved results on open-domain text-to-video generation using VQGAN-like framework. It shows impressive results when conditioning on a variable length of open domain texts. However, the paper misses important baselines and contains some unconvincing yet tempting conclusions. If the authors can resolve these concerns and add the corresponding experiments and discussions, I would like to change my current rating.

---

> ### Author Response · Authors · 2022-11-09
> **Authors' response**
>
> === Please check [**our common response**](https://openreview.net/forum?id=vOEXS39nOF&noteId=wruHGDsH7jV) for a newly added comparison with TATS and compute numbers.
>
> **3D Convolution versus C-ViViT:**
>
> As motivated in the introduction, our goal is to use video and image datasets at the same time. This means Phenaki should be able to encode single images and variable length videos. 3D Convolution in TATS and in general cannot do this out of the box. If you attempt to encode a video with padding except for the first image, the representation becomes corrupted and there is no guarantee that the representation is purely image representation. In C-ViViT, we make sure the extracted representation in the first image tokens are solely information from the first image via causal masking of time. This is a key requirement in our pipeline because we are interested in using image and video datasets together as data sources for learning. In addition, 3D convolution in TATS has not been shown to be applicable for video prediction for the previously mentioned reasons. This is a key requirement to be able to do text-image to video, text-video to video generation. There are established baselines for this such as Kinetics 600 and BAIR Robotpush that TATS does not include in their evaluations.
> Our paper already contains comparisons with previous 3D convolution based methods in our video prediction evaluation on Kinetics 600 and BAIR Robotpush (Table 4 against Video VQVAE and Table 5 against VideoGPT), and we outperform them. Both train an initial 3D convolution network to extract latent codes and train an additional model for video prediction on the latents. Having said that, in our common response to all reviewers we provide a direct comparison against TATS in a class conditional generation setting showing that even early in training we already outperform TATS by a significant margin.
>
> **Stylized videos:**
>
> Thank you for the question. To provide evidence that styles are learned from image datasets and not video datasets, we have updated our website (https://phenaki.video/style_videos.html) with a section showing that the models trained with 100% videos cannot stylize videos when the prompt specifies it. We provided the example with the pencil drawing panda and an additional water color style of the same panda and showed that the model trained with only video data cannot generate the right style because such examples simply don’t exist in the video dataset. However, the model learned these styles from the image dataset and the model trained jointly on images and videos is capable of generating such videos.
>
> **Variable length text prompts claim:**
>
> Thank you for the clarification. We agree with the reviewer on the distinction. “Open Domain” is already reflected in the title of the paper and  we updated the introduction to adjust this claim.
>
> **MaskGIT ablations:**
>
> Since we did not modify the original MaskGIT architecture, we found it to behave similarly to how it is reported to behave in the original MaskGIT paper. The only difference here is that we use more tokens because of our use of video data, but the behavior of all hyper parameters is essentially the same.
>
> To elaborate more on what the MaskGIT paper already states and as stated in our supplementary material, each video generation in our paper only requires 48 sampling steps while the autoregressive counterpart would take 1536 sampling steps. MaskGit provided a study of this in their paper (https://arxiv.org/pdf/2202.04200.pdf, Fig 4) that shows the advantage over autoregressive decoding. In addition, \lambda controls alignment with the input prompt and assuming the reviewer means maskgit temperature by \beta, it controls diversity of the generation. We point the reviewer to the MaskGIT paper for more details.

---

> > ### Comment · Reviewer_Qmqe · 2022-11-17
> > **Response**
> >
> > I appreciate the time and efforts made by the authors on the additional experiments, clarification of the raised confusions, and more qualitative results that have made the conclusions more sound. There is a clear advantage of using masked transformer over autoregressive transformer for video generation on the sampling efficiency, which is often a pain in generating videos. Therefore, I would love to increase my rating to reflect these. However, as also pointed out by R-E9Bs and R-2JKT, could the authors highlight the changes in the paper to reflect the responses?

---

> > > ### Author Response · Authors · 2022-11-17
> > > **RE: Response**
> > >
> > > We have highlighted the changes with blue color in the latest version of our manuscript, and addressed the comments by reviewers R-E9Bs and R-2JKT.

---

### Official Review · Reviewer_xbMS · 2022-10-25

**Confidence:** 5
**Correctness:** 3
**Technical Novelty And Significance:** 3
**Empirical Novelty And Significance:** 3
**Recommendation:** 6

**Clarity, Quality, Novelty And Reproducibility:**

### Clarity and Reproducibility
There is no problem with clarity and reproducibility because the proposed architecture is a straightforward extension of the existing architecture. If the checkpoints for the reported performance on experiments are released, the reproducibility of the works will be better.

### Quality
The quality of the paper can be improved better by explaining details of the imported architecture and training, sampling technique in the paper. How about adding preliminary parts for explaining those?

### Novelty
Novelty of this paper is clear. Although it borrows architectures from the several previous works, the combination itself is non-trivial and has a impact on video generation task.

**Strength And Weaknesses:**

### Strength
**appropriate modifications of previous works for video generation**

This paper modifies ViT-VQGAN and MaskGIT for video generation. These modifications seem to be plausible to process the video data. In the aspect of masked visual token modeling for video, this approach is fancy and new although implemented architectures are borrow from the previous works

**joint learning of image and video datasets**

This work demonstrate how to use both image and video datasets for video generation. I think that this idea is crucial for future research on video sythesis. I hope that there will be more analysis on this idea.

### Weaknesses

**validity of the statement (novelty over the ViViT architecture)**

I think that the descriptions in the last 4 sentences in **Novelty over the ViViT architecture** are wrong. The main statement here insists that causal attention is needed to support arbitrary temporal length. As I understand, C-ViViT architecture with all-to-all attention (I understand bidirectional attention) still supports encoding and decoding for arbitrary temporal length. (If it is not the case the bidirectional transformer can not model the masked token sequences.) If my conjecture is right, this work should discuss the difference between all-to-all and causal attention. For example, comparing the reconstruction performance differences between two attentions.

In this aspect, long video generation is not related to this video tokenizer, it is a matter of auto-regressive generation using bidirectional transformer inspired from MaskGIT. These confusing descriptions appear several times on the paper. Do I understand wrong about the proposed method?

**analysis on joint learning of image and video datasets**

It is not a weakness but suggestion. If possible, analyze the vocabularies for image and video dataset. For example, we can use this analysis to generate video with the text with out-of-vocaburary in video dataset and in-vocaburary in image dataset and check the quality of this video.

**Summary Of The Paper:**

This work introduces masked visual token modeling (MVTM) for video generation by extending ViT-VQGAN and MaskGIT into the video domain. The modified ViT-VQGAN tokenizer for video has two blocks with spatial- and temporal attention. To mitigate the nature of fixed code resolution in image (or video) tokenizer, it modifies full attention in the temporal blocks into causal attention. MaskGIT-like masked token modeling learns the distribution of videos represented as the sequence of tokens. The novelty of this work lies in the joint leaning on image and video datasets. Examples in the paper demonstrate that those combination generalize well on

**Summary Of The Review:**

This paper introduces a straightforward extension of masked visual token modeling for the video domain. By designing proper settings for video, the proposed method can generate high-quality, diverse-length videos from the various text description. The current version of the paper makes me confused about some details of the works, However, I believe that the overall work is quite fascinating for future research on video synthesis.

---

> ### Author Response · Authors · 2022-11-09
> **Authors' response**
>
> === Please check at [**our common response**](https://openreview.net/forum?id=vOEXS39nOF&noteId=wruHGDsH7jV) for a newly added comparison with TATS and compute numbers.
>
> **Validity of the statement (novelty over the ViViT architecture)**
>
> We thank the reviewer for the insightful question. While all-to-all attention does support arbitrary temporal length in the sense that you can drop future frames by simply dropping the attention on those frames as needed, the extracted representation becomes affected (will be different) because during training the model becomes dependent on the information provided by future frames. This is because the output of the self-attention is a weighted sum of all the value vectors that participate in the attention as stated in Vaswani et al., 2017. Therefore, when we re-encode the last n-frames from a previously generated video to continue the generation beyond the training length (only observe 1.4 seconds during training), the representation will be different from what is observed in training time because information from future frames is no-existent, and so, will cause semantic and visual discontinuities.
>
> One can argue that randomly providing different lengths of video to the all-to-all attention transformer is the same as applying causal attention, but there is no guarantee that the representation at each frame t will strictly depend on only information up to frame t. The causal attention in C-ViViT makes sure that the representation at frame t strictly depends only on information up to frame t.
>
> In addition, in this paper we are interested in learning from image and video datasets due to the lack of text-video datasets currently in the field. Therefore, C-ViViT always makes sure that the tokens of the first image are tokens that only depend on image features and nothing else. This enables our model to be optimized for text2video and text2image concurrently.
>
> **Analysis on joint learning of image and video datasets.**
>
> We would like to thank the reviewer for the suggestion. We have done this analysis in terms of different styles learned from image datasets which do not exist in video datasets. Please see an example of this here: https://phenaki.video/style_videos.html. We show how the model trained with only videos does not know about styles such as pencil drawing and water colors while the model trained with additional image datasets know about these styles.

---

> > ### Comment · Reviewer_xbMS · 2022-11-22
> > **Thank you for the comments**
> >
> > Thank you for the rebuttal to my questions and sorry for the late response.
> >
> > I can understand your points where I was confused. I've increased my confidence level to 5.
> >
> > By the way, I am still confused about the second paragraph in your comments "One can argue ~~ up to frame t". It is because the cited work [53] in the main paper already deals with video generation with all-to-all attention (+ sliding window fashion sampling) rather than causal attention. I think that my thought is not an important topic or argument for reviewing this work. However, I hope that the final version of the work discusses more related works, such as [53].

---

### Official Review · Reviewer_2JKT · 2022-10-28

**Confidence:** 4
**Correctness:** 3
**Technical Novelty And Significance:** 3
**Empirical Novelty And Significance:** 4
**Recommendation:** 8

**Clarity, Quality, Novelty And Reproducibility:**

### Clarity
- Overall, this paper is easy to read and understand. But the organization might be improved. I recommend using a subsection of evaluation metrics for experiments. Also what is the difference between FID video in Table 1 and FVD? CLIP means CLIP score?
- The reference style is different from the default ICLR style. Please check it.

### Novelty
C-ViViT is beyond a simple extension of ViViT and the authors proposes some interesting ideas.

### Reproducibility
Even if the authors explain some details of implementation and experimental configurations, it is not easy to reproduce because they did not present the detailed information on video data. Also, they did not submit the source code and write reproducibility section. So, reproducibility is restricted.



**Details Of Ethics Concerns:**

As well knowned, text-to-image or video generation can make a harmful contents given harmful text prompts. The authors properly disclaim this issue in Ethics statement.

**Strength And Weaknesses:**

### Strength
- Video generation from text prompts is a challenging and innovative research topic. In particular, time-variable video generation from text condition is a very meaningful achievement.
- The proposed method including C-ViViT and Transformer-based decoding seems a little novel. C-ViViT looks beyond simple extension of ViViT.
- The authors provide the results of five meaningful tasks, which are very promising and impressive.
- The paper is easy to read and clear.

### Weakness
Overall, I like this paper. But there are some room for improvements.
- [Major] Despite its impressive results, more ablation studies or analyses on key hyperparameters are required. For example, the number of video tokens seem have influence on the quality and compuational cost (both training and inference). The trade-off analysis is helpful. The results on maksing ratio and predicted token ratio can help to understand the model properties.
- [Major] In addition to quality metrics, the results on inference time can enhance the contributions.
- [Minor] How is the performance on few shot or fine-tuning of Phenaki in Table 1?
- [Minor] How long can the model consistently or stably generate videos for prompt sequences when generating time-variable generation?
- [Minor] Why did the author use 50%/50% setup for Table 1 instead of 80/20 setup even if 80/20 is better than 50/50?
- [Minor] TATS [Ge et al. 2022] needs to be added in related work. The comparison is not required because it is published in ECCV 2022. Also, because the authors present image conditional video generation, some video generation works need to be added in the related work such as StyleGAN-V [Skorokhodov et al. 2022] and DIGAN [Yu et al. 2022].

### References
- Ge et al. [Long Video Generation with Time-Agnostic VQGAN and Time-Sensitive Transformer](https://arxiv.org/abs/2204.03638). ECCV 2022.
- Skorokhodov et al. [StyleGAN-V: A Continuous Video Generator with the Price, Image Quality and Perks of StyleGAN2](https://arxiv.org/abs/2112.14683). CVPR 2022.
- Yu et al. [Generating Videos with Dynamics-aware Implicit Generative Adversarial Networks](https://arxiv.org/abs/2202.10571). ICLR 2022.


**Summary Of The Paper:**

This paper proposes a new video generation method from given text prompts, called Phenaki. The proposed method address four tasks such as text conditional video generation, text-image conditional video generation, time-variable text-conditioned video generation, video prediction, and image conditional video generation. For achieving this, the authors advanced ViViT to C-ViViT and designed Transformer-based decoder. They compare their method with multiple state-of-the art methods including NUWA, NUWA-Infinity, and CogVideo. The results look promising.


**Summary Of The Review:**

Overall, I like this paper. Although there are some room for improvement (please see the weakness), I decided to give accept as score considering the meaningful contributions of this paper.

---

> ### Author Response · Authors · 2022-11-09
> **Authors' response**
>
> === Please check at [**our common response**](https://openreview.net/forum?id=vOEXS39nOF&noteId=wruHGDsH7jV) for a newly added comparison with TATS and compute numbers.
>
>
> **Despite its impressive results, more ablation studies or analyses on key hyperparameters are required.**
>
> We thank the reviewer for this insightful question. The number of video tokens has different effects in different parts of our pipeline as described below:
>
> 1) Stage1 (C-ViViT) training. The number of tokens used during C-ViViT training affects the reconstruction quality. If we have less compression (i.e., more tokens), the reconstruction improves due to C-ViViT using more tokens to describe the input video. Therefore, less tokens results in worse reconstruction performance. However, as we see in our comparisons with the encoder-decoder baselines, our C-ViViT model is still better than the baselines at reconstructing video regardless of only using almost half the number of tokens of the baselines.
>
> 2) Stage2 (MaskGIT) training. The number of tokens has an effect on the maximum size of model we can fit during training. The more tokens we have, the smaller the model we can fit in memory due to the all-to-all attention in the MaskGIT transformer. As we have seen lately in the transformer literature, large models tend to perform better than smaller models.
>
> Therefore, we made a design choice of minimizing the number of tokens while maximizing the reconstruction performance between stage1 and stage2 models.
>
> Given the high training cost of the model, we did not do a thorough analysis on the masking ratio when training MaskGIT. We simply used the same masking ratio strategy used in the MaskGIT paper since they already did this study.
>
>
> **In addition to quality metrics, the results on inference time can enhance the contributions.**
>
> We provide the inference time to generate 1024 frames in our common response to all reviewers as well as newly added Section C.2 in the Appendix where we compare against TATS, and show we are faster.
>
>
> **How is the performance on few shot or fine-tuning of Phenaki in Table 1?**
>
> We have not had time to run these experiments, but we agree with the reviewer that it would be interesting to see this number.
>
>
> **How long can the model consistently or stably generate videos for prompt sequences when generating time-variable generation?**
>
> We don’t know for sure, but we have been able to generate up to 2.5 minutes so far. We have not really tried beyond that for time variable prompts. We did try to generate 5 minutes of just “First person view of parkour over buildings.” and the model continued generating sensible first person view of parkour.
>
> **Why did the author use 50%/50% setup for Table 1 instead of 80/20 setup even if 80/20 is better than 50/50?**
>
> Mostly due to the time and computational constraints. We started joint training using the 50/50 version and only later in the process figured that the 80/20 version works better. Unfortunately, we did not have the time nor the compute to reproduce the results of Table 1 with the new model and instead reported the numbers we had from an older version.

---

> > ### Comment · Reviewer_2JKT · 2022-11-13
> > **Thank you for the response**
> >
> > Thank you for your response on my questions.
> >
> > I agree with you that there is too short time to show more results.
> >
> > BTW, I cannot find the new compute time results you mentioned in the appendix. I found FVD result of TATS only there.
> > Could you share a link of 5 min generate video you mentioned or add it into your project page?
> >
> > Also, I recommend using other colored fonts for the revised parts so that reviewers can focus on those parts better.

---

> > > ### Author Response · Authors · 2022-11-17
> > > **RE: Thank you for the response**
> > >
> > > We have highlighted the changes with blue color in the latest version of our manuscript. We have added the computation requirements in the supplementary material, as well as links to our website showcasing two 5 minute videos. Here is the direct link to the videos https://phenaki.video/five_min_videos.html

---

### Official Review · Reviewer_E9Bs · 2022-11-01

**Confidence:** 5
**Correctness:** 3
**Technical Novelty And Significance:** 3
**Empirical Novelty And Significance:** 3
**Recommendation:** 6

**Clarity, Quality, Novelty And Reproducibility:**

- **Clarity**: The explanation of certain parts of the model are lacking and confusing. T5X is not elaborated on and I found myself referencing the original ViViT paper for illustrations and explanations since Figure 2 was unclear. What are the subscripts for and which dimension they operated over, etc.  It is unclear if the left C-ViViT illustration on figure 2 focuses on one spatial patch position over time? Additionally, the video generation process is unclear, specifically how decoding to the image space occurs for the different tasks demonstrated.
- **Originality**: Although the technical novelty of this work is low, the ingenius approach to combination of these tools for the purposes of video generation is novel and unique. It brings to video generation several capacities that have existed apriori on their own, but never in one model. Those being unconstrained long-horizon video generation, long form text conditional generation/prompting, fast generation, online video steering, video infilling/painting, and optional pre-training on image datasets. This combination of attributes do not exist in prior work making this article very significant.
- **Quality**: The engineering quality of this work is high, but the written article requires more work.

**Details Of Ethics Concerns:**

Although the models developed in this work are not publicly released. The datasets used to train the models are troublesome. An audit of one of these [datasets](https://arxiv.org/abs/2110.01963) raised significant ethical concerns and the authors have chosen to take the step of not open sourcing anything about their work until they have dealt with ensuing ethical issues. I argue that even though the trained models are better off not released, the code should still be open sourced to aid accurate comparison and benchmarking. Dealing with the issues of these models will take alot more time and much more effort than one team can muster.

**Strength And Weaknesses:**

## Strengths:
1. The chosen approach allows for unprecedented flexibility and performance in conditional video generation tasks.

2. It is a simple extension of existing architectures from other domains (video classification, image generation)

3. A good awareness of the ethical considerations with regard to this kind of research and its potential misuses.


## Weaknesses:
1.  **Technical Novelty**: The technical novelity in this article is quite limited. Phenaki is essentially a merger of MaskGIT and ViViT (and T5), the ensuing capabilities and advantages of this model come from the capabilities of the base models being used together. The technical novelity in the paper is limited to intelligent architectural choices. Of note is the addition of text conditioning to the bidirectional vision transformer and the change towards a causal transformer for better temporal modelling of video, allowing for unconstrained video generation.

2. **Code (and Demo) is not available**: The model itself is not dangerous but its uses could be. I would agree with keeping the trained models closed source since they could immediately be misused by the lay individual. But keeping the code closed source does not benefit the field as a whole and prevents researchers, reviewers and other academics from carrying out accurate experiments, bechmarks and comparisons. There are already several attempts at reproducing the model already. It would be interesting to see what the ethical reviewers have to say with regard to this argument. For the sake of accurate benchmarking and comparisons, I would urge the authors to open source their code since the stated reasons for keeping it closed source (i.e. "understanding of data, prompt and output filtering") are not impacted by this.

3. **Clarity**: Figure 2 is incomplete and the explanation of encoder-decoder model unclear.  It can be argued that the decoding process is one of the most important procedures in this work yet it is barely covered. It is unclear whether decoding and generation primarily happens in the latent space? are the outputs from previous timesteps decoded and then re-encoded and fed into the bidirectional transformer? can the entire decoding process for several timesteps purely occur in the latent space?

4. **Missing Sections** : The computational requirements and cost of training such models is of increasing interest to the wider community. There are references throughout the paper to computational cost and efficient but this topic is never fully explored. It would be good to get some idea of computational requirements, from the equipment used for the experiments to (tpu/gpu hrs/days, flips), to wall-clock inference speed and time (per frame, time-step, etc). This is not in the article or its associated supplementary material.

5. **T5X**: There is mention of discussing each of the model components in the paper in the last line of the first paragraph in section 2. But T5X is not elaborated on. What is T5X? Is this the original T5 model repackaged? Is it the T5-XXL (4.6B) model?

6. **LAION**: Mentions of the dangers of the LAION-400M dataset are provided without reference. The work that raised this issues should be directly referenced within that paragraph (ie. [Multimodal datasets](https://arxiv.org/abs/2110.01963)).

7. **UCF-101** : The standard UCF-101 benchmark is missing and should be included. It is a tractable and accessible benchmark that allows for better comparison to prior art within the video generation field. I encourage the authors to have a look at Table 1.d in the [TATS](https://arxiv.org/abs/2204.03638) paper. The model presented in this work should be evaluated on base level video generation ability and class-conditional video generation capability. Whether the text prompt is a sentence or a word/action should not present a challenge but such benchmarks are useful for evaluating base level capabilities of such models. In the case of using pre-trained models, it should be highlighted that the model is pre-trained on a range of other datasets rather than the standard practice of training on only the benchmark dataset. It would also be useful to know of its base-level long horizon [video prediction](https://wilson1yan.github.io/teco/) capabilities and dynamics.



### Missing References:
The field of video generation has existed for some time now and was long established within the GAN world. References to this prior art are missing. These include but are not limited to:

- [Long Videos of Dynamic Scenes](https://arxiv.org/abs/2206.03429)
- [RV-GAN](https://openaccess.thecvf.com/content/CVPR2022W/WiCV/html/Gupta_RV-GAN_Recurrent_GAN_for_Unconditional_Video_Generation_CVPRW_2022_paper.html) - CVPR-W 2022
- [TATS](https://songweige.github.io/projects/tats/index.html) - ECCV 2022
- [StyleGAN-V](https://openaccess.thecvf.com/content/CVPR2022/html/Skorokhodov_StyleGAN-V_A_Continuous_Video_Generator_With_the_Price_Image_Quality_CVPR_2022_paper.html) - CVPR 2022
- [LDVD-GAN](https://www.sciencedirect.com/science/article/pii/S0893608020303397) - Neural Networks 2020
- [DIGAN](https://arxiv.org/abs/2202.10571)

I would recommend the authors have a look at Section 2, paragraph 2, of the [DiGAN](https://arxiv.org/abs/2202.10571) paper covering prior video generation work.

**Concurrent work** such as [Imagen](https://arxiv.org/abs/2210.02303) and [Make-a-video](https://arxiv.org/abs/2209.14792) should also be refenced in the revised article.


**Summary Of The Paper:**

This article presents, Phenaki, an approach to text-to-video modelling and synthesis based on an autoregresserve transfomer-based encoder-decoder model.

# Stage 1. C-ViViT:
Frames from a given video are first linearly projected into patch-embeddings then processed by a spatial transformer; the resulting output of the spatial transformer is further processed across time with a causal (temporal) transformer.

# Stage 2. MaskGIT
The outputs of the causal transformer are project to a discrete latent space via a learned codebook. A bidirectional transformer is trained via a self-supervised reconstruction mechanism (masking) to reconstruct sequences of the tokens output by the causal transformer. This is done by taking the discretized outputs of stage 1 and randomly masking (according to some procedure) parts/subsets of these causal tokens. The bidirectional transformer is trained to undo the masking operation and reconstruct the original unmasked token sequence. The bidirectional transformer also conditions this reconstruction process on text information (ie. prompt) provided by consuming embeddings provided by a pretrained language model.

# Stage 3: Text driven Video Generation
Once all these pieces come together, generation occurs via providing empty/blank tokens and a text prompt to the bidirectional transformer. The bidirectional transformer outputs a set of tokens that align with those in the codebook, these tokens are projected back to the image space by an image decoder that is the inverse of the ViViT procedure and architecture. Auto-regressive generation can be achieved by shifting the tokens generated by the bidirectional transformer forward by one time-step, appending an empty/blank token for the new time step and providing that to the bidirectional transformer along with the text prompt.


**Summary Of The Review:**

Overall this is a great piece of work and a significant contribution to the field. If the authors remedy the issues i raised as weaknesses then I will increase my score. But as is, I support the publication of this work at ICLR.

---

> ### Author Response · Authors · 2022-11-09
> **Authors' response**
>
> === Please check at [**our common response**](https://openreview.net/forum?id=vOEXS39nOF&noteId=wruHGDsH7jV) for a newly added comparison with TATS and compute numbers.
>
> **Technical novelty**
>
> As mentioned by other reviewers, the novelty of this paper is in:
> 1) The combination of different modules. Similar to Reviewer E9BS which mentions that “the ingenious approach to combination of these tools for the purposes of video generation is novel and unique” and also Reviewer xbMS: “the combination itself is non-trivial and has an impact on video generation task.” we also believe that the combination is not trivial while being novel.
> 2) C-ViViT: This is a novel architecture that is a big step forward building on top of the previous work. As mentioned by Reviewer Qmqe: “The architecture and video results are novel.” Please note that C-ViViT and its novel architecture is the main building block that enables Phenaki to generate consistent videos for which can span over multiple minutes.
> 3) The task. To the best of our knowledge, this is **the first** paper that addresses generating videos from open-domain stories. The task in itself is novel and important. We believe moving forward the majority of text-to-video papers will try to support story driven video generation which makes this paper even more impactful.
>
> **Code (and demo) is not available**
>
> We are actively working towards making the code available as soon as practically feasible. Due to a number of constraints (e.g. resources) we are not able to provide a demo at this time, but we hope that once the code is available, anyone can train their version of this model on publicly available data..
> However, we should mention that the paper provides detailed descriptions of architecture and hyper-parameters used to enable other researchers to reproduce our work. There are already a few implementations on github.
>
> **Clarity**
>
> To clarify, the video generation happens in latent space given text for the first M frames (1.4 seconds in our experiments) only. The subsequent frames are generated by re-encoding the last N frames (.75 seconds in our experiments) where N < M using C-ViViT and generating the next F frames conditioned on the previous N frames and text.
>
> Our method cannot do generation purely in latent space due to the design choices we made to take advantage of text-image and text-video datasets during training. The first I tokens are purposely single image (space only) tokens and the next V tokens are video (space-time) tokens. Therefore, MaskGIT will always expect this input arrangement and if we unroll MaskGIT in latent space by a sliding window approach, the first tokens will become video tokens and not image tokens anymore, effectively breaking the required input arrangement.
>
> **Missing sections:**
>
> We added two new sections in the appendix to address this issue. First, we have provided a comparison against a previous method (TATS) on class conditional video generation on UCF-101 in Section C.1 of the Appendix. Second, we provided training and inference computation numbers in Section C.2 of the appendix.
>
> **T5X**
>
> We use T5-XXL to pre-compute text representations used by our model. We updated the text to clarify this.
>
> **LAION-400M**
>
> Thank you for pointing us to the work in Multimodal Datasets. We added the reference when discussing the dangers of LAION-400M.
>
> **UCF-101**
>
> We are currently running experiments on UCF-101 from scratch (not from pretrained models), and provide the numbers in the common response to all reviewers..
>
> **Missing references:**
>
> Thank you for pointing out these references. We added them  in the revised version.

---

> > ### Comment · Reviewer_E9Bs · 2022-11-15
> > **Highlight Changes in revised manucript**
> >
> > It would be good to highlight the recommended changes from the different reviewers in the manuscript with different color codes. Its difficult to decipher whether all recommended changes were made.
> >
> >
> > The explanation of video generation posted by the authors under the clarity heading in the above comment should also be in the main paper. This is an important point.
> >
> > References to prior art are still missing.
> >
> > It seems that a few of the recommendations made by myself and the other reviewers have not been made as of yet. My score for this article remains unchanged.

---

> > > ### Author Response · Authors · 2022-11-17
> > > **RE: Highlight Changes in revised manucript**
> > >
> > > We have highlighted the changes with blue color. We cite all the works suggested by the reviewers, provided the clarification about auto-regressive video continuation, T5-XXL clarification, citation about LAION, and UCF-101 experiments in the most recent version of our manuscript.

---

> > > > ### Comment · Reviewer_E9Bs · 2022-11-23
> > > > **Missing citations**
> > > >
> > > > Again, you claim to have cited all the works suggested by the reviewers.
> > > > And this is the third time you have said so while missing references..
> > > > Could you go through my comments and make sure of this before claiming so again..
> > > >
> > > > Most importantly, concurrent work such as [Make-a-Video](https://arxiv.org/abs/2209.14792) is still missing.

---

> > > > > ### Author Response · Authors · 2022-11-23
> > > > > **Missing references**
> > > > >
> > > > > Can you please be more specific on which ones are missing? We checked the list and it seems they are all there (except for the parallel work). Citing the parallel work which is also a submission to this venue might be against ICLR rules as they both came out after the deadline. Make-A-Video doesn't cite this paper either. But we are happy to cite it as a parallel work if the AC is OK with it.

---

### Author Response · Authors · 2022-11-09
**Comparison with TATS and video generation runtime**

# Quantitative comparison with TATS:

We would like to thank the reviewers for pointing us to TATS by Ge et al. 2022. To compare against TATS in an established dataset for video generation, we conducted an experiment on class conditional video generation on the UCF-101 dataset as requested by the reviewers. Our results so far (currently 540K iterations through training) are as follows:


| Method | FVD (lower is better) |
| -----------| ----------------------------|
| TATS | 332 |
| Phenaki (345M) | 250 |


We expect this gap to get larger by the time Phenaki is done training. We should mention that given the time constraints of the rebuttal, we are training a smaller version of our model with 345M parameter (in contrast to the best model in the paper which contains 1.8B parameters). We expect results to be better with a larger model.

# Running time for generating a video:
Since TATS provides a generation time for a 1024-frame video, we measure the time it takes our biggest model (with 1.8B parameters) to generate the same number of frame. Please note that these numbers might not be directly comparable since the accelerator used to generate the videos is most likely different.

| Method | Time to generate 1024 frames |
| ---------- | ---------------------------------------- |
| TATS-base | 30min |
| TATS-hierarchical | 7.5min |
| Phenaki (1.8B) | 4.1min |

We updated the appendix of the paper to include both of these tables.

---

> ### Public Comment · ~Eugene_Lee1 · 2022-11-11
> **Runtime of model in the paper**
>
> We couldn't find the runtime of the model found in the submitted paper. The only source of runtime is specified here and it's only for the larger (1.8B) model. It'd be helpful if the runtime of the smaller model is also added to the supplementary paper for reference as the demo images and videos of Phenaki corresponds to the results of the smaller model mentioned in the paper if my understanding is correct. This information would be helpful for future researchers that are interested in this topic but have limited computational resources.

---

> > ### Author Response · Authors · 2022-11-11
> > **Model used in text-video experiments.**
> >
> > Thanks for your question. As specified in the first sentence of the second paragraph in Section 3.1, "Unless specified otherwise, we train a 1.8B parameter Phenaki model". In other words, the model used for images and videos in the paper/website come from the 1.8B model, our best model.

---

> > > ### Public Comment · ~gao_kai_hang1 · 2023-03-12
> > > **早餐店的老板问我要吃什么**
> > >
> > > 早餐店的老板问我要吃什么

---

> ### Comment · Reviewer_E9Bs · 2022-11-15
> **Second table (Running time for generating a video) missing?**
>
> Sorry, but where is the second table listed in the appendix or paper?

---

### Decision · Program_Chairs · 2023-01-20

**Decision:**

Accept: poster

**Justification For Why Not Higher Score:**

The writing quality and paper clarity can be further improved. More ablation study on the hyper-parameters of MaskGIT and some other aspects of the model can be added. Since code and pre-trained data cannot be released for now, the reproducibility can be restricted.

**Justification For Why Not Lower Score:**

All the reviewers recommend accept, the AC also thinks that the paper makes valuable contributions, there is no reason to reject the paper.

**Metareview: Summary, Strengths And Weaknesses:**

This paper introduces Phenaki, a new encoder-decoder model for open-domain text-to-video generation based on the proposed C-ViViT and MaskGIT training.

Initially, this paper received mixed scores. After author rebuttal, the scores have changed to 6668. All the reviewers are generally happy about the paper, and recommend acceptance. Specifically, all the reviewers agree that (1) the method is novel and unique, and (2) results are impressive with unprecedented flexibility and performance in conditional video generation tasks. During rebuttal, the authors have further provided additional comparison with TATS by Ge et al. 2022. On the other hand, the writing quality and paper clarity can be further improved, more ablation on the hyper-parameters of MaskGIT can be added, and reproducibility is restricted.

All reviewers agree to accept the paper, and the AC also thinks that this paper makes solid contributions, and would like to recommend acceptance of the paper.

**Note From Pc:**

if the above contains the word "oral" or "spotlight" please see: "oral" presentation means -> notable-top-5% and "spotlight" means -> notable-top-25%. As stated in our emails, we are disassociating presentation type from AC recommendations